

# *Endozoicomonas* dominance and Vibrionaceae stability underpin resilience in urban coral *Madracis auretenra*

Jordan Ruiz-Toquica[1,2], Andrés Franco Herrera[2] and Mónica Medina[3]

[1] Corporation Center of Excellence in Marine Sciences-CEMarin, Bogota, Colombia
[2] Academic Area of Environmental and Biological Sciences, Jorge Tadeo Lozano University, Santa Marta, Magdalena, Colombia
[3] Department of Biology, The Pennsylvania State University, State College, PA, United States of America

## ABSTRACT

Coral resilience varies across species, with some exhibiting remarkable stability and adaptability, often mediated by their associated microbiomes. Given the species-specific nature of coral-microbiome interactions, investigating the microbiomes of urban-adapted corals provides critical insights into the health, dynamics, and functioning of coral holobionts. In this study, we examined the microbiome of *Madracis auretenra*, a Caribbean coral from Santa Marta, Colombia, across contrasting environmental conditions. Over two years, we compared the microbiomes of healthy and stressed coral colonies from two distinct reef habitats—urban and protected—using 16S rRNA gene sequencing (V4 region) to asses microbial diversity. Our findings revealed microbial richness and diversity were primarily influenced by seasonal and local factors rather than host-specific traits such as interaction with algae, health status, or microhabitat. These variations were not substantial enough to disrupt the overall microbial community structure, which remained stable across temporal and spatial scales. Dominant taxa included *Endozoicomonas*, along with Vibrionaceae and Rhodobacteraceae, which form dense ecological interaction networks. Notably, nutrient and oxygen levels emerged as key drivers of microbiome fluctuations, yet Vibrionaceae populations exhibited exceptional temporal stability. These findings highlight the presence of a well-structured and resilient coral microbiome with minimal seasonal variability, even in urban-influenced environments. We propose that the dominance of *Endozoicomonas* and the stability of Vibrionaceae populations play a pivotal role in maintaining microbiome balance, ultimately contributing to the ecological resilience of *M. auretenra* in dynamic reef habitats.

# INTRODUCTION

Coral reefs, among the most biodiverse ecosystems on Earth, are increasingly threatened by anthropogenic stressors such as climate change, pollution, and habitat degradation (*Heron et al., 2016*; *Hughes et al., 2017*; *Li et al., 2023*). These stressors disrupt coral health, reducing resilience and accelerating reef decline (*Eddy et al., 2021*; *Hoegh-Guldberg & Bruno, 2010*;

Corresponding authors
Jordan Ruiz-Toquica,
jordan.ruiz@utadeo.edu.co
Mónica Medina, mum55@psu.edu,
momedinamunoz@gmail.com

*Wenger et al., 2016*). However, the resilience of coral reefs is not uniform—various coral species display remarkable adaptability, often attributed to the critical role of their microbial communities, collectively known as the coral microbiome (*Mohamed et al., 2023*; *Peixoto et al., 2020*; *Voolstra et al., 2024*). Urban coral reefs—those near densely populated coastal urbanized areas—offer a unique lens to study this adaptability. Despite exposure to intense sedimentation, eutrophication, and pollution (*Burt et al., 2020*; *Heery et al., 2018*), urban corals sustain stable and functionally diverse microbiomes that support holobiont health and resilience (*Farias et al., 2023*; *Wainwright et al., 2019*). This suggests that these unique microbial communities are integral to their capacity to withstand environmental stressors.

The coral microbiome, primarily composed of bacteria, archaea, fungi, and viruses, plays a crucial role in nutrient cycling, pathogen defense, and stress mitigation (*Bourne, Morrow & Webster, 2016*; *Krediet et al., 2013*; *Peixoto et al., 2017*). Healthy corals maintain balanced microbiomes, whereas stressed individuals—such as those experiencing bleaching or diseases—undergo microbial shifts that increase microbiome heterogeneity (*Boilard et al., 2020*; *Gong et al., 2020*; *MacKnight et al., 2021*). These disruptions, often compromising holobiont homeostasis and exacerbating coral vulnerability, are driven by environmental fluctuations, including temperature and nutrient levels (*Dunphy, Vollmer & Gouhier, 2021*; *Wang et al., 2018*; *Zaneveld et al., 2016*; *Zaneveld, McMinds & Thurber, 2017*). Nutrient pollution enhances microbial richness, while elevated temperatures diminish beneficial microbes and favor opportunists (*Maher et al., 2019*; *Maher et al., 2020*; *Webster et al., 2016*). Seasonal variations, including rainy periods, further influence microbiome diversity (*Bulan et al., 2018*; *Paulino et al., 2023*; *Yu et al., 2021*). Competition and algae proximity also alter microbiome composition leading to a significant decline in diversity (*Briggs, Brown & Osenberg, 2021*; *Brown, Lipp & Osenberg, 2019*). To cope with this environmental stress, coral microbiomes employ two primary strategies: stability and flexibility.

Corals with stable microbiomes retain core microbial assemblages, preserving essential ecological functions under stress (*Dunphy et al., 2019*; *Epstein, Torda & Van Oppen, 2019*; *Hadaidi et al., 2017*). In contrast, corals with flexible microbiomes undergo greater compositional shifts, facilitating rapid adaptation to fluctuating conditions (*Maher et al., 2020*; *Röthig et al., 2020*). For instance, some corals adjust their microbial populations along anthropogenic gradients while maintaining core community structure (*Delgadillo-Ordoñez et al., 2022*; *Ziegler et al., 2016*; *Ziegler et al., 2019*), whereas others exhibit distinct microbiomes across locations, underscoring compositional flexibility (*Ricci et al., 2022*; *Tandon et al., 2022*). Both strategies allow corals to optimize their microbiomes by recruiting beneficial microbes and suppressing pathogens, enhancing coral resilience and acclimatization to environmental stress (*Montaño Salazar, Quintanilla & Sánchez, 2023*; *Reshef et al., 2006*; *Ritchie, 2006*). These adaptive mechanisms are particularly pronounced in urban coral microbiomes.

Urban coral microbiomes differ significantly from those in less-impacted habitats. Corals in Florida reefs exhibit increased stress-tolerant bacterial groups during thermal events (*Rosales et al., 2019*). In Singapore, coral microbiomes display high spatial variability over small scales, reflecting plasticity to respond to localized stressors (*Wainwright et al., 2019*). Similarly, corals along Colombia's urbanized coasts harbor site-specific microbial

communities (*Cárdenas et al., 2012*; *Roitman et al., 2020*; *Ruiz-Toquica et al., 2023*), and corals in Hainan Island, South China Sea, undergo seasonal microbiome shifts, possibly making them more resilient to deteriorating conditions (*Zhu et al., 2023*). Yet, research on urban coral microbiomes remains scarce. These microbiomes often harbor taxa capable of pollutant degradation, heavy metal sequestration, and nutrient tolerance, suggesting functional adaptations to urbanized environments (*Rosenberg et al., 2022*). As urban reefs become increasingly prevalent, understanding their microbiomes is essential for uncovering mechanisms underpinning coral resilience in highly impacted habitats.

This study examines the microbiome of *Madracis auretenra* in the urbanized reefs of Santa Marta Bay, Colombia, a region experiencing high sedimentation, nutrient enrichment, and habitat degradation. Our previous research indicates that *M. auretenra* maintains a "good" health index despite these challenges, suggesting a resilient and adaptive microbiome (*Ruiz-Toquica et al., 2023*), that serves as a model of the response of the coral microbiome under urban influence. To better understand this resilience, we investigated microbiome dynamics across two contrasting locations, examining its response to environmental factors, health status, and algal interactions over two years. Our findings offer key insights into urban coral microbiomes and their role in enhancing coral holobiont resilience under anthropogenic pressure.

## MATERIALS & METHODS

### Permits

The collection was conducted under authorized permits for the non-commercial scientific study of wild species, in compliance with Colombian environmental regulations. Permit PIR 007-2021, issued through Resolution 191 on December 16, 2021, by the Ministry of Environment and Sustainable Development and National Natural Parks of Colombia, governed this specific collection. The framework permit was granted under Resolution 1715 (December 30, 2015) and amended by Instruction Normative No. 00213 (January 28, 2021), as issued by the National Environmental Licensing Authority (ANLA).

### Sample collection and study design

Samples were collected from two locations in the Colombian Caribbean: a protected site, Chengue Bay (CHE) (11°19′30″N, 74°07′41.8″W), situated within the Tayrona National Natural Park (Tayrona NNP), and an urban site, Inca Inca Beach (INC) (11°12′58.4″N, 74°14′6.3″W), influenced by urban settlements in Santa Marta Bay (Fig. 1A). Chengue Bay represents a non-urban, conserved environment, while the urban site at Inca Inca Beach has shown significant benthic community degradation (*Ruiz-Toquica et al., 2023*). Despite these differences, both locations harbor similar-sized reef patches of the coral *Madracis auretenra*, enabling comparative microbiome analysis across environmental gradients.

The study spanned two years (2022–2023) and encompassed dry (cooler waters, 20°–25 °C, salinity >36) and rainy seasons (warmer waters, 27°–29 °C, salinity <34). Collections were conducted at seasonal peaks: March and October 2022, and April and November 2023. At each site, a 30 m$^2$ permanent transect belt was established within *M. auretenra* patches to facilitate repeated sampling. Individual coral colonies, separated by >5 m to

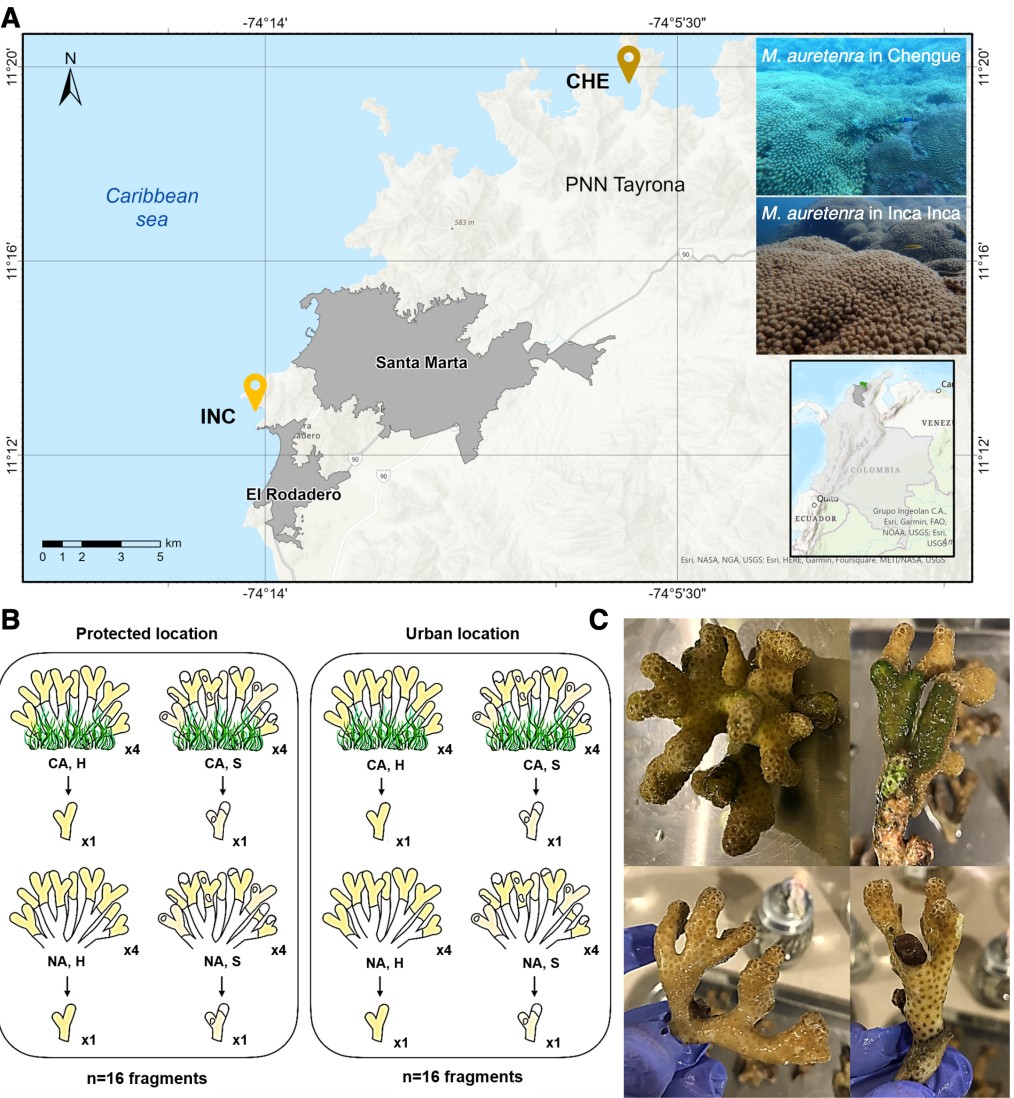

**Figure 1 Study area and sample collection design.** (A) Two sampling locations: Chengue Bay (CHE) (11°19′30″N, 74°07′41.8″W), now referred to as the protected site, and the Inca Inca beach (INC) (11°12′58.4″N, 74°14′6.3″W), now referred to as the urban site. (B) Hierarchical sampling design: CA, fragments in contact with algae (physically touching); NA, fragments with no contact; H, healthy fragments; and S, stressed fragments. (C) Pictures of healthy fragments (top) in contact with algae (right) and with no contact (left), and stressed fragments (bottom) in contact with algae (right) and with no contact (left).

minimize pseudoreplication, were randomly selected within the transect. Due to colony morphology and branch overlap, tagging was not feasible; thus, randomly chosen branches were used for all samples.

At each time point, we identified four donor colonies per site based on health status (healthy or stressed) and interaction with algae ("in contact" and "non-contact") (Fig. 1B). Healthy colonies were visually identified by their bright yellow to golden coloration and absence of deterioration, while stressed colonies exhibited signs such as bleaching, paling,

or partial mortality (*Gil-Agudelo et al., 2009*) (Fig. 1C). Colonies classified as "in contact" were those with algae (*e.g.,* turf, crustose, or fleshy) visibly touching the coral tissue, whereas "non-contact" colonies had no algal interference (Fig. 1C). One branch was detached per colony (a total of 128 branches for the entire study).

To minimize microbiome variations related to tissue maturity branches approximately five cm in length were collected, avoiding distal tips (juvenile tissue) and basal segments (senescent tissue). We used new sterile gloves for each sample to prevent cross-contamination and carefully detached and placed them in individual sterile Ziplock bags according to their condition. Samples were transported on ice to maintain integrity during the one-hour transit. Additionally, two liters of seawater were collected in triplicate along the transect belt, approximately five cm above the coral colonies. These samples were transported in refrigerated sterile bottles for immediate microbiome composition and physicochemical analysis.

## Sample processing and DNA extraction

Samples were processed immediately at the Molecular Ecology Laboratory, Universidad de Bogotá Jorge Tadeo Lozano, Santa Marta, following protocols adapted from *Cárdenas et al. (2012)* and *Ruiz-Toquica et al. (2023)*. To standardize the starting material, coral branches were cut into three cm fragments and washed five times with sterile phosphate-buffered saline (PBS; 137 mM NaCl, 2.7 mM KCl, 100 mM $Na_2HPO_4$, 2 mM $KH_2PO_4$, pH 7.4) to remove loosely associated bacteria. The four washed fragments per condition were pooled and placed in flasks containing 40 mL of sterile PBS.

Coral mucus was extracted by stirring the samples at 135 rpm for 1 h at 28 °C. Tissue slurries were obtained by macerating the remaining fragments and recovering the suspension in 40 mL of PBS. Both mucus and tissue samples were centrifuged at 3,000 $g \times 15$ min, and the resulting pellets were resuspended in six mL of sterile PBS and stored at −20 °C. For seawater samples, one L from each replicate was filtered through 0.22 µm Millex®-GS syringe filters (Merck Millipore). Filter membranes were detached using sterile pruners, transferred to sterile two mL tubes, and stored at −20 °C until further processing.

To prevent sample degradation, total DNA was immediately extracted using the DNeasy® PowerSoil® Pro Kit (Qiagen), following the manufacturer's protocol with modifications. Coral samples (500 µL of mucus/tissue slurries) and seawater filter membranes were combined with 800 µL of Solution CD1 in PowerBead Pro tubes. The tubes underwent bead beating at maximum speed for 5 min, followed by a 2-min pause and incubation at 4 °C, repeated twice. After this disruption step, the manufacturer's protocol was followed.

DNA was eluted in 80 µL of elution buffer, and quality was assessed *via* electrophoresis on 0.8% agarose gels in 0.5X TBE buffer (45 mM Tris-borate, 45 mM boric acid, 1 mM EDTA, pH 8.2) and quantified using a Nanodrop Multiskan™ Go Microplate. To enhance the recovery of low-abundance microbial DNA, extractions were performed in triplicate, pooled, and stored at −20 °C for downstream analyses.

## Library preparation, sequencing, and bioinformatic processing

All samples underwent simultaneous library preparation and sequencing. The 16S rRNA V4 hypervariable region was amplified using a modified primer set: CS1_515F (5′-ACA CTG ACG ACA TGG TTC TAC AGT GCC AGC MGC CGC GGT AA-3′) and CS2_806R (5′-TAC GGT AGC AGA GAC TTG GTC TGG ACT ACH VGG GTW TCT AAT-3′) (*Roitman et al., 2020*). These primers include common sequence linkers at the 5′end and Illumina MiSeq adapters for tagging and indexing (*Caporaso et al., 2012*). PCR reactions used 50 ng of high-quality DNA in a 50 μL reaction mix containing: 5 μL 10X PCR Buffer II (Quantabio), 2.5 μL dNTPs (10 mM), 2.5 μL of each primer (10 μM), 1.5 μL MgCl$_2$ (50 mM), 0.2 μL AccuStart II™ Taq Polymerase (5 U μL$^{-1}$), and molecular-grade water.

Amplification was performed with an initial denaturation at 94 °C for 2 min, followed by 35 cycles of 94 °C for 45 s, 55 °C for 1 min, and 72 °C for 1.5 min, with a final extension at 72 °C for 7 min. Amplicons (~250 bp) were verified by electrophoresis on 2% agarose gels; no contamination was detected in negative controls. PCR products were stored at −20 °C until further processing. The barcoding and sequencing steps were carried out on the Illumina MiSeq platform at the DNA Services Facility, University of Illinois at Chicago, USA. Both positive and negative controls were included.

Demultiplexed paired-end reads were processed with the DADA2 pipeline (https://benjjneb.github.io/dada2/tutorial_1_8.html, accessed January 31, 2024) in RStudio (v. 4.3.2) (*Callahan et al., 2016*). Reads were truncated at 220 bp (forward) and 200 bp (reverse) based on quality profiles, filtered with an expected error threshold of 2 (*Prodan et al., 2020*), and merged to generate a denoised table of Amplicon Sequence Variants (ASV). Chimeric sequences were removed, and taxonomy was assigned to the genus level using the SILVA prokaryotic SSU database v138.1. The ASV identified as mitochondria, chloroplasts, or present in the negative controls (*e.g., Ralstonia* and *Escherichia-Shigella*) were removed following decontamination protocols (*Karstens et al., 2019*). Further taxonomy and prevalence filtering excluded singletons, unassigned taxa (NA), and phyla with prevalence below the 5% threshold. Rarefaction curves confirmed sequencing depth sufficiently captured microbial diversity. Data were normalized to a sequencing depth of 1,771 reads per sample using the *rarefy_even_depth* function in the Phyloseq package (v. 1.46.0) (*McMurdie & Holmes, 2013*).

## Diversity analysis

The normalized phyloseq object was used to assess alpha and beta diversity across samples. Observed richness, Shannon, and Gini-Simpson (1-D) diversity indices were calculated and expressed as mean ± standard error (SE). Statistical differences in alpha diversity were analyzed across factors, including site, season by year, algae contact condition, health status, and compartment (mucus, tissue, and seawater). Levene's test was used to evaluate the equality of variances, while differences in alpha diversity were tested using the Kruskal–Wallis test. Significant results ($p < 0.05$) were further analyzed through *post-hoc* pairwise comparisons using Tukey's HSD (Honestly Significant Difference) test.

Beta diversity was analyzed to detect differences among sites, seasons by year, and compartments using Permutational Multivariate Analysis of Variance (PERMANOVA)
with 1,000 permutations and Bray-Curtis distance metrics, as implemented in the VEGAN package (v.2.6-4) (*Oksanen et al., 2018*). Pairwise PERMANOVA tests were conducted for significant results, with *p*-values adjusted using the Benjamini–Hochberg (BH) correction to mitigate Type I error (*Benjamini & Hochberg, 1995*). Beta dispersion (PERMDISP) assessed the homogeneity of group dispersion, while an Analysis of Similarity (ANOSIM) was used to compare Bray-Curtis distances among groups (*Khomich et al., 2021*). Microbial community structures were visualized *via* non-metric multidimensional scaling (NMDS).

## Taxonomic analysis and differential abundance

Taxa were agglomerated at rank levels, and read counts were converted to relative abundances (%). Shared and not-shared taxa among microbiomes under different conditions were visualized using weighted Venn diagrams *via* the *ps_venn* function in the MicEco package (v.0.9.19). The core microbiome was identified using the *core* function in the Microbiome package (v.1.24.0) (*Wen et al., 2023*) and was defined as microbial taxa present in at least 50% of samples with a relative abundance threshold of 0.2%, representing taxa consistently shared within the host microbial community (*Neu, Allen & Roy, 2021*).

A differential abundance analysis was performed using the DESeq2 package (v.1.42.0) (*Nearing et al., 2022*). Unrarefied filtered ASV tables were agglomerated at the family level. Features with more than 90% zero values were excluded. Pairwise comparisons were conducted, with size factors estimated using the *poscounts* method to account for zero inflation. Adjusted *p*-values were calculated using the BH correction following fitting to negative binomial generalized linear models with the Wald *post-hoc* test. Significant taxa ($p < 0.05$) were visualized in heatmaps generated using the Complex Heatmap package (v.2.18.0).

An analysis of the composition of microbiomes with bias correction (ANCOM-BC) was performed to additionally evaluate significant changes in bacterial family relative abundances using the ANCOMBC package (v.2.4.0) (*Lin & Peddada, 2020*). Log fold changes, standard errors, statistical tests, and *p*-values were computed *via* the *ancombc* function, with adjustments using the Holm-Bonferroni method with *p*-values <0.05 indicating differential abundance.

## Correlation of physicochemical factors with coral microbiome and co-occurrence network analysis

Seawater samples were processed following (*Ruiz-Toquica et al., 2023*). Three liters of each seawater sample ($n = 8$) was analyzed by the LABCAM unit of the Marine and Coastal Research Institute-Invemar for nitrites ($NO_2^-$), nitrates ($NO_3^-$), ammonia ($NH_4^+$), orthophosphates ($PO_4^{3-}$), and total suspended solids (TSS) using the *Strickland & Parsons (1972)* methods. The results were accredited by the "Instituto de Hidrología, Meteorología y Estudios Ambientales-IDEAM" under Resolution 0049 of 15 January 2021. Three measurements of temperature, pH, salinity, and oxygen saturation were taken *in situ*. Data were expressed as mean ± standard error (SE), and a log ($x + 1$) transformation and Euclidean distance matrix were applied to prepare data for statistical testing. Significant differences were determined using PERMANOVA in PRIMER7 (*Clarke & Gorley, 2006*),

with "site" and "season" as factors. Unrestricted permutations of raw data (999 iterations) were used to account for type III errors. A principal component analysis (PCA) and a similarity percentage analysis (SIMPER) were conducted to explore relationships between variables and identify their contributions.

Correlations between the coral microbiome attributes and the environmental variables were assessed using a canonical correspondence analysis (CCA) with the *envfit* function in the vegan package (v.2.6–4) (*Oksanen et al., 2018*). Filtered ASV tables were scaled to unit variance. The variance inflation factor (VIF) was calculated using *vif.cca*, excluding variables with VIF >10. Significant variables ($p < 0.05$) were used to construct *envfit* arrows, with arrow length indicating the correlation coefficient with the ordination axis. Environmental factors were plotted alongside sample data ordination, and the most abundant active ASV were aggregated at the phylum level for the correlation analysis. The *anova.cca* (9999 permutations) function tested significant correlations ($p < 0.05$) between mean ASV abundance and environmental factors. Kendall rank correlation coefficients were calculated for the most abundant taxa and environmental variables, with *p*-values adjusted using the Benjamini–Hochberg method. Significant correlations (adjusted $p < 0.05$) were visualized using a heatmap generated with *ggplot* from the ggplot2 package (v.3.5.0).

To explore potential ecological interactions among microbial taxa, co-occurrence networks were constructed using the Sparse Inverse Covariance Estimation for Ecological Association Inference (SpiecEasi) package (v.1.1.3) (*Kurtz et al., 2015*). A centered log-ratio transformation was applied to the data, followed by sparsity estimation through the Stability Approach to Regularization Selection (StARS) model. The Meinshausen-Bühlmann neighborhood selection method was used with a variability threshold of $10^{-2}$ (*Liu, Roeder & Wasserman, 2010*). Networks were plotted at the family level using the *plot_network* function.

## Deposits and data availability

Three specimens (three branches from independent colonies) of the coral *Madracis auretenra* from each location were deposited at the Marine Museum of Natural History of Colombia-Makuriwa, part of the José Benito Vives de Andréis Marine and Coastal Research Institute (INVEMAR). Deposits comply with the Single Regulatory Decree of the Environmental Sector 1,076 of 2015 (which consolidates Decrees 1,375 and 1,376 of 2013) and Andean Nations Decision 391 of 1996. Collection details were documented in the Information System on Marine Biodiversity (SiBM). Specimens are cataloged in the Cnidarian Catalogues under accession numbers INV CNI4625 to INV CNI4630 and consecutive numbers 80041 to 80046.

Raw sequencing data have been deposited in the National Center for Biotechnology Information (NCBI) under BioProject accession PRJNA1103847. Sample-specific information is available under BioSample accessions SAMN41063004 to SAMN41063075, and fastq files are archived in the Sequence Read Archive (SRA) under accession numbers SRR28786279 to SRR28786350. The R scripts used for the DADA2 pipeline, statistical analyses, figure generation, and filtered tables of ASV, taxonomy, metadata,

and environmental data are available in a public GitHub repository: https://github.com/
JorMicrobe/Madracis_auretenra_Microbiome.git.

## RESULTS

### Microbial diversity and community structure in *Madracis auretenra* across temporal and spatial scales

We obtained a total of 2,703,519 sequencing reads across 72 samples, with 1,426,194 high-quality reads retained after filtering steps (Table S1). This dataset yielded 7,875 amplicon sequence variants (ASV), identified as single nucleotide variants. Sequencing depth ranged from 2,023 to 46,724 reads per sample (Fig. S1).

The microbiome richness of *M. auretenra* varied significantly by "site" (Kruskal–Wallis; $Chi^2_{(1,71)} = 6.94$, $p = 0.008$), and "season" (Kruskal–Wallis; $Chi^2_{(3,69)} = 31.67$, $p = 0.000$). However, no significant differences were observed between healthy and stressed corals, corals "in contact" *versus* "non-contact" with algae, and between compartments (mucus and tissue) ($p > 0.05$) (Table S2). Similar trends were noted for the Shannon diversity across sites (Kruskal–Wallis; $Chi^2_{(1,71)} = 12.35$, $p = 0.000$), seasons (Kruskal–Wallis; $Chi^2_{(3,69)} = 22.98$, $p = 0.000$), and coral compartments (Kruskal–Wallis; $Chi^2_{(2,70)} = 13.63$, $p = 0.001$), reflecting local, temporal and spatial alpha diversity patterns. Overall, microbial diversity was higher at the urban site (Fig. 2A), and in the dry season of 2022 (Fig. 2B). In comparison, the seawater microbiome exhibited greater diversity overall relative to coral compartments (Fig. 2C, Table S3).

Taxonomic overlap analysis revealed that most ASVs were shared across sites (97%) (Fig. 2A), while only 44% between seasons (Fig. 2B), suggesting taxonomic consistency across spatial rather than temporal gradients. Coral tissue and mucus shared 50% of the taxa but exhibited minimal overlap with the surrounding seawater (10%). Notably, coral mucus did not harbor not-shared taxa, whereas seawater exhibited 24% of not-shared ASV (Fig. 2C), suggesting unique taxa for coral tissue and the seawater microbiomes.

Conversely, microbial community structure showed no significant differences across sites (PERMANOVA; $pseudo\text{-}F_{(1,71)} = 2.31$, adjusted-$p = 0.167$, $R^2 = 0.032$), seasons (PERMANOVA; $pseudo\text{-}F_{(3,69)} = 2.53$, adjusted-$p = 0.252$, $R^2 = 0.100$), and coral compartments (PERMANOVA; $pseudo\text{-}F_{(2,70)} = 48.93$, adjusted-$p = 0.074$, $R^2 = 0.586$) indicating consistent beta diversity across temporal and spatial scales (Table S4). This was further supported by ANOSIM analysis, which suggested weak variation among groups ($0.25 < R < 0.5$) (Fig. 3A) and homogeneity in group dispersion, as indicated by PERMDISP (sites: $p = 0.756$, $F = 0.945$; seasons: $p = 0.27$, $F = 0.938$; compartments: $p = 0.108$, $F = 0.853$). In contrast, the microbiome of *M. auretenra* was significantly distinct from the seawater microbiome (PERMANOVA; $pseudo\text{-}F_{(2,70)} = 48.93$, adjusted-$p = 0.001$, $R^2 = 0.586$) (Fig. 3A, Table S4). These findings underscore the unique microbial associations of *M. auretenra* and their stability across spatial and temporal scales.

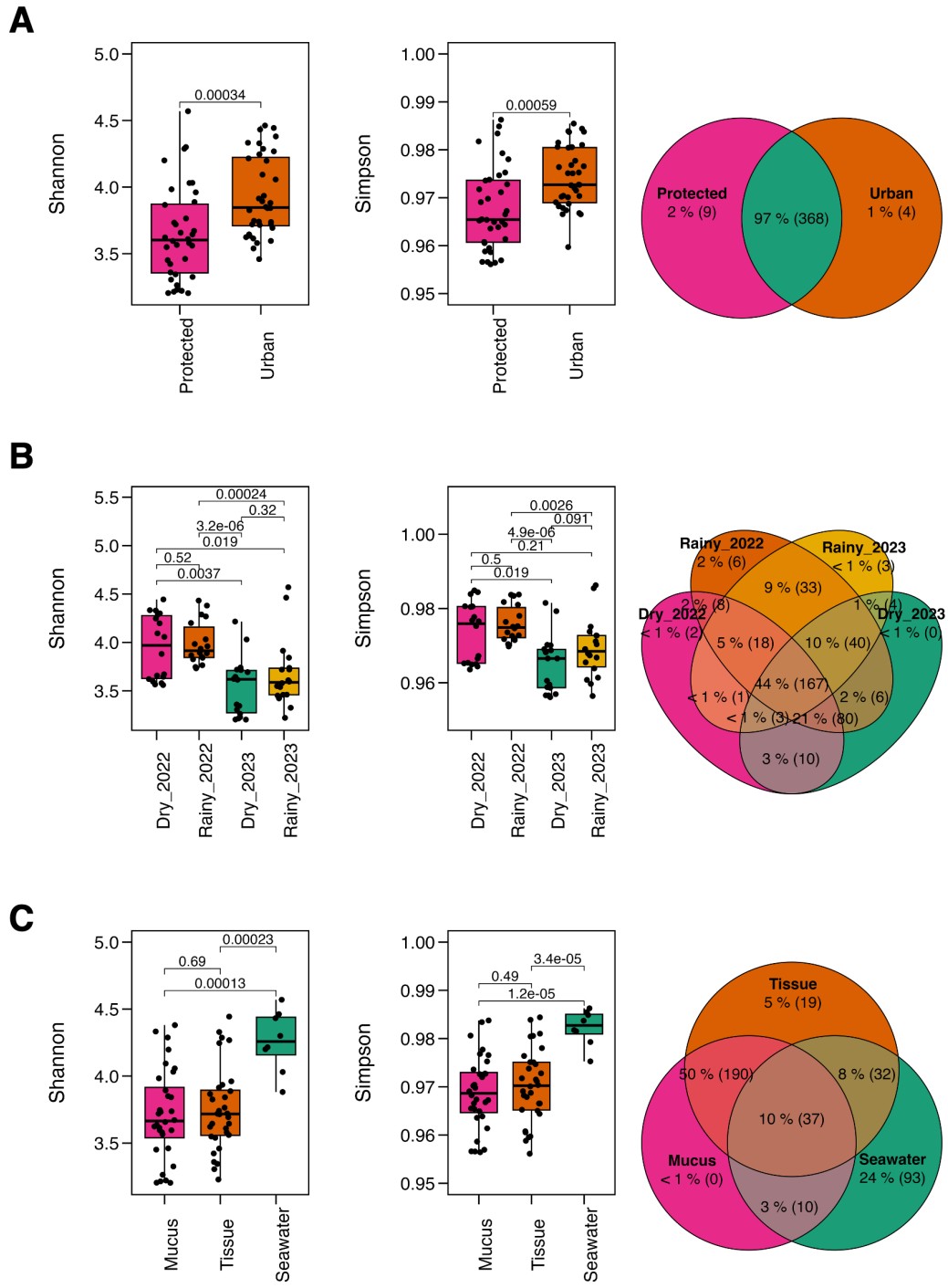

**Figure 2  Alpha diversity metrics.** The Shannon diversity index and Gini-Simpson diversity index along a Venn diagram illustrating the overlap of shared and not-shared ASVs, categorized by (A) site (protected *vs.* urban), (B) season (dry *vs.* rainy, 2022–2023), and (C) compartment (coral mucus, coral tissue, and surrounding seawater). *P*-values above each comparison represent the results of the Kruskal–Wallis test, with significance defined as $p < 0.05$.

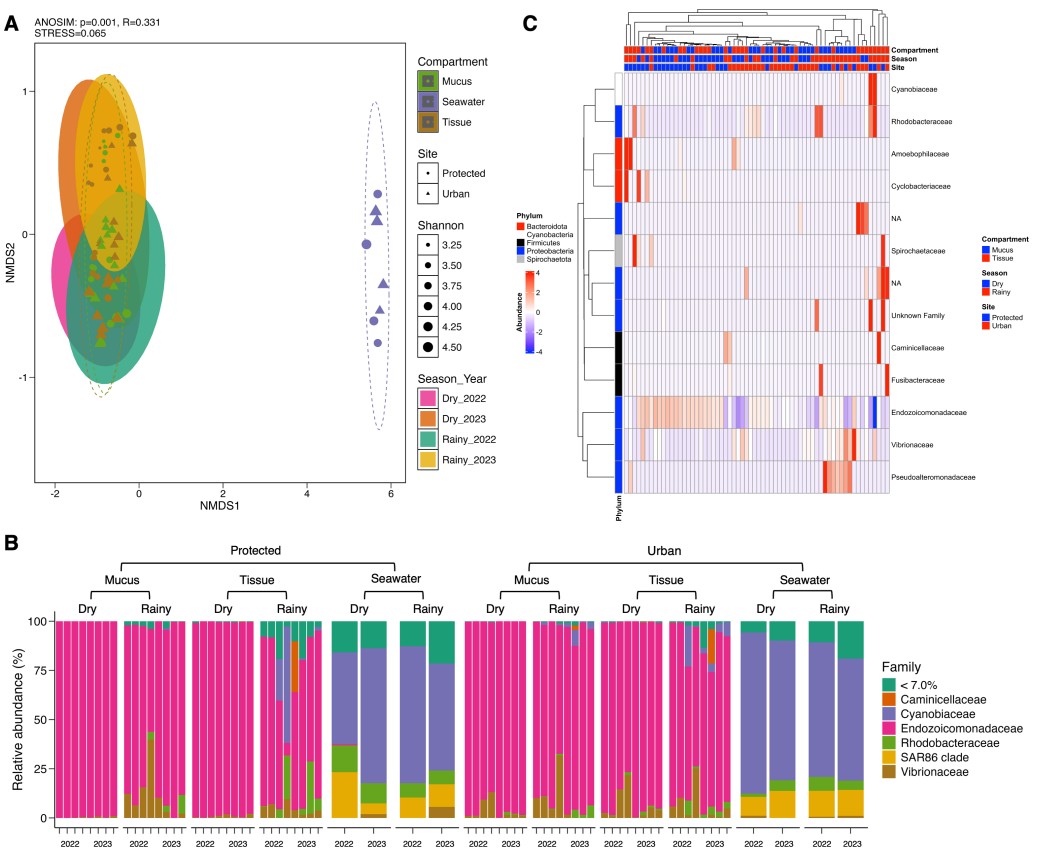

**Figure 3** Structure and composition of the microbiomes associated with coral compartments (mucus and tissue) and the surrounding seawater. (A) Non-metric multidimensional scaling (nMDS) ordination showing differences between the coral microbiome and the surrounding seawater microbiome (ANOSIM, $p = 0.001$, $R = 0.331$), and temporal similarities in the coral microbiome structure over time (PERMANOVA; pseudo-$F_{(3,69)} = 2.53$, adjusted- $p = 0.252$, $R^2 = 0.100$) and space (PERMANOVA; pseudo- $F_{(1,71)} = 2.31$, adjusted- $p = 0.167$, $R^2 = 0.032$). The symbols represent the sites, and the symbol colors the compartment (mucus, tissue, seawater). (B) Coral and seawater microbiome composition. Relative abundance (%) at the family level is distributed among coral compartments (mucus and tissue) and seawater samples. The eleven most abundant families are plotted, and the remaining taxa are grouped under abundance <7.0%. (C) Differential abundance analysis using DESeq2. The heatmap colors show the differential relative abundance by clustering the top eleven families by site, season, health status, contact with algae, and compartment. NA are not assigned families. The red color indicates an increase in relative abundance, while the blue color indicates a significant decrease (GLM; $p = 0.000$). White color indicates the absence of this taxon in the given sample (relative abundance = 0).

## Dominant and core taxa of *Madracis auretenra* and seawater microbiomes

The microbiome of *M. auretenra* ($n = 64$) was predominantly composed of members of the families Endozoicomonadaceae ($88.14 \pm 1.99\%$), Vibrionaceae ($4.97 \pm 0.97\%$), Cyanobiaceae ($2.22 \pm 1.03\%$), and Rhodobacteraceae ($1.93 \pm 0.57\%$) (Fig. 3B). In contrast, the seawater microbiome ($n = 8$) was dominated by Cyanobiaceae ($65.37 \pm 3.84\%$), SAR86 clade ($12.54 \pm 1.80\%$), Rhodobacteraceae ($7.11 \pm 1.27\%$), Flavobacteriaceae ($4.94 \pm 0.82\%$), and Vibrionaceae ($1.28 \pm 0.66\%$) (Fig. 3B).

Within the coral microbiome, the relative abundance of Endozoicomonadaceae (genus *Endozoicomonas*) was highest in both mucus and tissue, followed by Vibrionaceae (Fig. 3B, Table S5). Seasonal fluctuations were notable, with *Endozoicomonas* abundance decreasing during the rainy season, particularly at the protected site (from 98.74 ± 0.28% to 64.57 ± 9.38%) (Fig. 3B, Table S6). This decline coincided with increased abundances of the families Cyanobiaceae, Rhodobacteraceae, and Vibrionaceae (Fig. 3B).

At the urban site, the abundance of Vibrionaceae in *M. auretenra* remained stable across seasons (~6.20%), but it was lower compared to the protected site (11.20 ± 4.50%). Among the less abundant families, Rhodobacteraceae and Cyanobiaceae were notably present across the samples, while Fusibacteraceae and Spirochaetaceae were exclusive to coral tissue during the rainy season (Fig. 3B). Abundant genera in coral compartments included *Catenococcus*, *Vibrio*, *Ruegeria*, *Cyanobium*, and *Synechococcus* while rare taxa included *Candidatus Amoebophilus* and *Paramaledivibacter* (Fig. S2).

The core microbiome was defined as taxa with a prevalence >50% and relative abundance >0.2%. A total of 47 ASV were identified as core members, predominantly from the families Endozoicomonadaceae, Vibrionaceae, and Rhodobacteraceae (Fig. S3). ASV within the genus *Endozoicomonas* (*e.g.,* asv0004, asv0012, asv0010, asv0001–3, and asv0006) were the most prevalent, with ~90% prevalence across samples (Fig. S4). Notably, only *Endozoicomonas* asv0010 was detected in both coral and seawater samples. Other highly prevalent taxa included *Catenococcus, Vibrio,* and *Ruegeria* (Fig. S3).

In seawater, Cyanobiaceae dominated in samples from both sites and across rainy seasons (Tables S6 and S7). Taxa from Flavobacteriaceae, Rhodobacteraceae, and the SAR86 clade were less abundant at the protected site compared to the urban site (Fig. 3B, Tables S6 and S7). The relative abundance of Vibrionaceae was similar across sites but increased during rainy seasons (Fig. 3B). The most abundant seawater genera included *Synechococcus*, followed by *Cyanobium*, the strain HIMB11, and the NS2b marine group (Fig. S2).

## Influence of seasonality, coral health, and coral-algae contact on the microbiome dynamics

Differential abundance analysis using DESeq2 and ANCOM-BC revealed significant shifts in the relative abundance of 13 bacterial families within the coral microbiome. Among these, the family Endozoicomonadaceae exhibited pronounced seasonality, with significantly lower relative abundance during rainy seasons (GLM; $p = 0.000$) (Fig. 3C, Table S8). No differential abundance of Endozoicomonadaceae was observed for location, coral-algae contact, or coral health status (Table S8). Interestingly, the family Vibrionaceae showed no significant changes in relative abundance across any of the conditions (Table S8). In contrast, Rhodobacteraceae, particularly the genus *Ruegeria*, displayed significantly higher abundance in stressed corals (GLM; $p = 0.000$) and in corals "in contact" with algae (GLM; $p = 0.000$) (Fig. S5). This family was not influenced by season or location (Table S8).

Family Cyanobiaceae demonstrated significant differential abundance across sites (GLM; $p = 0.01$) and seasons (GLM; $p = 0.01$) (Fig. 3C). Pseudoalteromonadaceae, although

present at low relative abundance (<2.5%), showed higher abundance in coral mucus (GLM; $p = 0.01$) and during the rainy season (GLM; $p = 0.01$) (Table S8). Other families, including Fusibacteraceae, Spirochaetaceae, Amoebophilaceae, and Caminicellaceae, exhibited condition-specific patterns, such as higher abundances in coral tissue, during rainy seasons, or at the protected site (GLM; $p < 0.05$) (Fig. 3C, Table S8).

**Correlation with environmental factors and ecological interactions in the microbial community**

Twenty-four seawater samples were analyzed for physicochemical parameters (Table 1 and Table S9). Nutrient levels displayed heterogeneous patterns: ammonia, nitrites, and phosphates remained consistent across sites and seasons, while nitrates and total suspended solids (TSS) differed significantly (Tables S10 and S11). The seasonal variation (PERMANOVA; $pseudo\text{-}F_{(3,23)} = 25.18$, $p = 0.001$) explained most of the change with key contributors, including temperature, salinity, and nitrates (Fig. S6). During dry periods, ammonia, nitrates, nitrites, and pH were dominant factors (>24% of contribution), whereas rainy periods were more influenced by TSS, phosphates, ammonia, and nitrates (>38%) (Table S12). The correlation analysis showed that ammonia, phosphates, nitrates, nitrites, and TSS explained over 66% of the microbial community variation (ANOVA-like permutational test; $F_{(5,66)} = 1.566$, $p = 0.016$) (Tables S13 and S14). These factors influenced phyla Proteobacteria, Firmicutes, and Thermoplasmatota, while nitrites predominantly shaped phyla Cyanobacteria, Actinobacteriota, and Bacteroidota (Fig. 4A). *Endozoicomonas* abundance positively correlated with low temperature and high phosphates (Kendall; 99.95%) and negatively with high TSS (<–99.95%) (Fig. 4B). Vibrionaceae abundance was associated with elevated TSS, nitrates, and phosphates, while Cyanobiaceae with high TSS and phosphates (Fig. S7).

Ecological interactions inferred from 381 ASV, using sparse inverse covariance (sparsity = 0.024, path length = 20), revealed three clusters: coral-associated taxa (Endozoicomonadaceae, Vibrionaceae, Rhodobacteraceae, and Pseudoalteromonadaceae), seawater-associated taxa (Cyanobiaceae, Actinomarinaceae, Flavobacteriaceae, and SAR86 clade), and a mixed group including Endozoicomonadaceae and Cyanobiaceae. The network topology suggests a mixture of tightly interconnected clusters and more peripheral nodes (Fig. S8), where Vibrionaceae, Endozoicomonadaceae, and Pseudoalteromonadaceae form a highly interconnected dense hub in the middle of the network, suggesting strong positive interactions between these taxa. Peripheral nodes with few connections indicate fewer interactions, particularly with seawater-associated taxa and the Alteromonadaceae family (Fig. S8).

## DISCUSSION

Microbial community profiles often appear consistent across coral taxa (*Ainsworth et al., 2015*; *Hernandez-Agreda et al., 2018*; *Lawler et al., 2016*). However, evidence of host specificity highlights distinct patterns, such as those observed in urban corals (*Ziegler et al., 2019*). This study investigated the microbiome of coral *Madracis auretenra* from Santa Marta Bay, Colombia, over two years. Microbial dynamics were compared between urban

Ruiz-Toquica et al. (2025), *PeerJ*, DOI 10.7717/peerj.19226

**Table 1  Values of the physicochemical variables of seawater during the dry and rainy seasons of 2022 and 2023 at the two locations.** Data are expressed as the mean ± standard error (SE). Temperature (°C), salinity, pH, oxygen saturation (%), nutrients: ammonia $NH_4^+$ ($\mu g\ L^{-1}$), orthophosphate $PO_4^{3-}$ ($\mu g\ L^{-1}$), nitrate $NO_{3-}$ ($\mu g\ L^{-1}$), and nitrite $NO_{2-}$ ($\mu g\ L^{-1}$), and total suspended solids (TSS) (mg $L^{-1}$) were measured for three independent replicas for each season ($n = 24$).

| Season-Year | Protected site Chengue Bay (Tayrona NNP) 11°19′30″N; 74°07′41.8″W | | | | Urban site Inca Inca 11°12′58.4″N; 74°14′6.3″W | | | |
|---|---|---|---|---|---|---|---|---|
| | Dry-2022 | Rainy-2022 | Dry-2023 | Rainy-2023 | Dry-2022 | Rainy-2022 | Dry-2023 | Rainy-2023 |
| Temperature (°C) | 26.37 ± 0.09 | 28.27 ± 0.03 | 26.08 ± 0.17 | 30.20 ± 0.12 | 26.30 ± 0.21 | 28.43 ± 0.15 | 26.33 ± 0.07 | 30.17 ± 0.03 |
| Salinity | 36.20 ± 0.06 | 29.73 ± 0.17 | 35.56 ± 0.14 | 29.90 ± 0.32 | 36.27 ± 0.09 | 29.80 ± 0.06 | 35.43 ± 0.09 | 31.03 ± 0.17 |
| pH | 8.57 ± 0.21 | 8.43 ± 0.01 | 8.12 ± 0.00 | 8.60 ± 0.00 | 8.36 ± 0.02 | 8.39 ± 0.05 | 8.13 ± 0.06 | 8.66 ± 0.05 |
| Oxygen saturation (%) | 73.03 ± 3.47 | 56.30 ± 0.70 | 73.50 ± 0.00 | 92.53 ± 10.86 | 69.90 ± 5.93 | 57.10 ± 0.69 | 85.90 ± 1.59 | 108.77 ± 10.92 |
| Ammonia $NH_4^+$ ($\mu g\ L^{-1}$) | 10.53 ± 0.53 | 10.00 ± 0.00 | 10.47 ± 0.47 | 11.10 ± 1.10 | 12.27 ± 0.57 | 10.00 ± 0.00 | 10.00 ± 0.00 | 10.00 ± 0.00 |
| Orthophosphate $PO_4^{3-}$ ($\mu g\ L^{-1}$) | 2.00 ± 0.00 | 3.76 ± 0.47 | 3.68 ± 0.48 | 2.00 ± 0.00 | 3.07 ± 0.13 | 4.53 ± 0.02 | 2.82 ± 0.57 | 2.00 ± 0.00 |
| Nitrate $NO_3^-$ ($\mu g\ L^{-1}$) | 4.98 ± 1.52 | 11.83 ± 0.41 | 5.68 ± 0.46 | 20.37 ± 6.43 | 21.12 ± 0.53 | 11.19 ± 1.31 | 2.10 ± 0.00 | 6.15 ± 3.54 |
| Nitrite $NO_2^-$ ($\mu g\ L^{-1}$) | 1.32 ± 0.21 | 0.70 ± 0.00 | 1.43 ± 0.06 | 0.79 ± 0.09 | 1.54 ± 0.06 | 0.70 ± 0.00 | 0.70 ± 0.00 | 0.70 ± 0.00 |
| TSS (mg $L^{-1}$) | 3.62 ± 0.04 | 11.43 ± 0.38 | 14.37 ± 0.84 | 4.06 ± 0.10 | 1.60 ± 0.11 | 23.87 ± 5.76 | 19.10 ± 4.25 | 3.32 ± 0.03 |

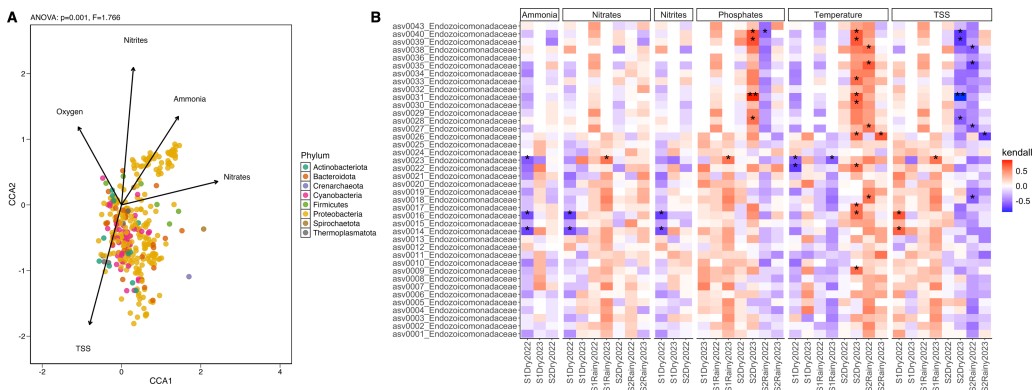

**Figure 4** **Differential abundance and the influence of the environmental conditions.** (A) Canonical Correlation Analysis (CCA) of the physicochemical variables shaping the structure of the microbiome. Ammonia (ANOVA-like permutational test; $F_{(1,66)} = 0.468$, $p = 0.001$), nitrates ($F = 1.022$, $p = 0.001$), nitrites ($F = 1.136$, $p = 0.001$), oxygen saturation ($F = 0.916$, $p = 0.001$), and TSS ($F = 0.656$, $p = 0.001$) explained more than 65% of the variation in the structure of the microbiome by phylum (CCA1: eigenvalue = 0.089, proportion = 36.82%; CCA2: eigenvalue = 0.075, proportion = 30.90%). (B) Heatmap of the Kendall rank correlation coefficient analysis. Axis $X$ corresponds to samples, and axis $Y$ to taxa (ASVs). Significance was detected with adjusted $p$-values less than 0.05 (*) and less than 0.01 (**). Red color indicates a strong positive correlation, whereas blue color indicates a strong negative correlation of the forty most abundant taxa.

and protected reef habitats and analyzed based on coral health, coral-algae interaction, and compartments. Results revealed a microbiome dominated by *Endozoicomonas* and a stable population of Vibrionaceae, influenced by seasonal weather patterns. Despite environmental variability, the microbiome demonstrated remarkable stability, underscoring its potential role in coral resilience under urban influence. A detailed analysis of the microbiome dynamics is presented below.

## The microbiome associated with *M. auretenra* demonstrates temporal and spatial stability

Coral-associated microbial communities are shaped by fluctuations in their microhabitat and the surrounding seawater, which can influence microbial composition, abundance, and diversity (*Gong et al., 2020*; *Marchioro et al., 2020*). Despite this dynamic nature, some coral microbiomes demonstrate remarkable stability under environmental variability (*Baquiran et al., 2025*; *Epstein, Torda & Van Oppen, 2019*). In our study, the microbiome of coral *Madracis auretenra* exhibited consistent community structure across spatial and temporal scales, remaining stable despite seasonal shifts in seawater parameters. Microbial richness and diversity patterns were primarily influenced by seasonal and spatial factors rather than host-specific traits, such as interaction with algae, health status, or microhabitat conditions. However, these variations were not substantial enough to disrupt community stability. Instead, the observed fluctuations in composition suggest corals can selectively acquire and shed microbial associates to enhance their response to environmental changes (*Reshef et al., 2006*; *Ricci et al., 2022*; *Tandon et al., 2022*). This dynamic microbial assembly process aligns with findings in other coral species, where shifts in microbiome composition
**Table 2 Summary of different studies that aimed to describe microbiome dynamics of different coral species.** The coral model, the assessed condition, the microbial assessment approach, the sample size and rarefaction, the number of ASVs, the diversity alpha metric (Shannon index), and the differentially abundant taxa at the family level are shown and compared with the present study.

| Study | Coral model | Condition | Microbial assessment | Sample size and rarefaction (reads sample$^{-1}$) | Number of ASVs | Diversity alpha (Shannon) | Differential abundant taxa (Family) |
|---|---|---|---|---|---|---|---|
| Leite et al. (2018) | Mussismilia hispida | Season (dry vs rainy) | V4 region 16S rRNA amplicon sequencing-Illumina | n=48 1,900 reads | Not reported | Higher in the rainy season | Peptostreptococcaceae, Comamonadaceae, Pseudomonadaceae and Flavobacteriaceae |
| | | Locality (river-influenced vs protected) | | | | Higher in the river-influenced locations | Moraxellaceae, Comamonadaceae |
| Epstein et al. (2019) | Acropora hyacinthus | Season (dry vs rainy) | V5–V6 region 16S rRNA amplicon sequencing-Illumina | n=216 | 14,083 | Higher in the rainy season | Endozoicomonadaceae, Burkholderiaceae |
| | A. spathulata | | | | | Higher in the dry season | Endozoicomonadaceae, Burkholderiaceae, and Methylophilaceae |
| Brown, Lipp & Osenberg (2019) | Porites sp. | Algae-interaction (algae-contact vs no contact) | V4 region 16S rRNA amplicon sequencing-Illumina | n=52 11,629 | 1,500 | No differences detected | Endozoicomonadaceae, Flavobacteriaceae, Bdellovibrionaceae, Piscirickettsiaceae, Rhodobacteraceae, and Fusibacteraceae |
| Rajasabapathy et al. (2020) | A. cytherea | Health status (healthy vs diseased) | Full-length 16S rRNA amplicon sequencing-Nanopore MinION | n=? | 1,623 | Higher in healthy mucus | Bacillaceae, Enterobacteraceae, Clostridiaceae, Caulobacteraceae, and Pseudomonadaceae |
| Marchioro et al. (2020) | A. tenuis | Compartment (mucus vs tissue) | V1–V3 region 16S rRNA amplicon sequencing-Illumina | n=136 3,500 reads | 12,051 | Higher in mucus | Synechococcaceae, Rhodobacteraceae, Verrucomicrobiaceae, Halomonadaceae, and Alteromonadaceae |
| | A. millepora | | | | | Higher in mucus | Endozoicomonadaceae, Synechococcaceae, Pirellulaceae, and Burkholderiaceae |
| Clements et al. (2020) | A. millepora | Algae-interaction (algae-contact vs no contact) | V3–V4 region 16S rRNA amplicon sequencing-Illumina | n=? 4,273 reads | Not reported | No differences detected | Oceanospirillaceae, Endozoicomonadaceae, and Vibrionaceae |
| Paulino et al. (2023) | Siderastrea stellata | Season (dry vs rainy) | V4 region 16S rRNA amplicon sequencing-Illumina | n=? | 31,645 | Higher in the dry season | Not assessed |
| | | Locality (river-influenced vs protected) | | | | Higher in the river-influenced location | Not assessed |
| | | Health status (healthy vs bleached) | | | | No differences detected | Rhodobacteraceae, Stappiaceae, Rhizobiaceae, Nitrospiraceae, and Leptospiraceae |
| Lu et al. (2022) | Platygyra sp. | Algae-interaction (algae-contact vs no contact) | V3–V4 region 16S rRNA amplicon sequencing-Illumina | n=27 1,230 reads | 1,526 | No differences detected | Saprospiraceae |
| | Montipora sp. | | | | | Higher in the interaction (algae-touching) | Rhodobacteraceae |
| | Pocillopora sp. | | | | | Higher in the interaction (algae-touching) | Rhodobacteraceae, and Alteromonadaceae |
| Present | Madracis auretenra | Season (dry vs rainy) | V4 region 16S rRNA amplicon sequencing-Illumina | n=72 1,771 reads | 7,785 | Higher in the dry season | Endozoicomonadaceae, Pseudoalteromonadaceae, and Cyanobiaceae |
| | | Locality (urban-influenced vs protected) | | | | Higher in the urban-influenced location | Cyanobiaceae, Spirochaetaceae, Amoebophilaceae, and Cyclobacteriaceae |
| | | Algae-interaction (algae-contact vs no contact) | | | | No differences detected | Rhodobacteraceae |
| | | Health status (healthy vs stressed) | | | | No differences detected | Rhodobacteraceae |
| | | Compartment (mucus vs tissue) | | | | No differences detected | Endozoicomonadaceae, Pseudoalteromonadaceae, Fusibacteraceae, Spirochaetaceae, Caminicellaceae, and Cyclobacteriaceae |

are driven by seasonal cycles, local environmental factors, habitat characteristics, and proximity to anthropogenic activities (summarized in Table 2). Such patterns underscore the intricate balance between microbial stability and flexibility, which is critical for coral resilience in fluctuating environments.

Stable microbiomes exhibit minimal variation over time, even under stress, reflecting co-evolutionary dynamics where microbes enhance coral fitness (*Dunphy et al., 2019*; *Dunphy, Vollmer & Gouhier, 2021*; *Hadaidi et al., 2017*). These robust microbiomes, dominated by specific taxa, adapt to environmental shifts through incremental adjustments to microbial composition while preserving their core structure (*Farias et al., 2023*; *Lima et al., 2023*; *Roder et al., 2015*). We identify two key mechanisms driving microbiome dynamics: microbiome reshaping, involving minor taxa shifts without altering the core structure (*e.g., M. auretenra* in urban-influenced Colombian Caribbean habitats), and microbiome restructuring, marked by substantial community reorganization and the emergence of a new microbial profile. The balance and interplay between these mechanisms likely underpin coral resilience and success in stressed environments, meriting further study.

These concepts extend to compartmentalization within the coral holobiont. Coral mucus often exhibits greater microbial diversity than other compartments (*e.g.,* tissue and skeleton) (*Bourne & Munn, 2005*; *Marchioro et al., 2020*), although contrasting patterns have been noted, particularly in the Caribbean where coral tissue is more diverse (*Pollock et al., 2018*). Flexible microbiomes show dynamic responses in mucus, reflecting heightened environmental influence (*Bhagwat, Ravindran & Irudayarajan, 2024*; *Brown & Bythell, 2005*; *Ricci et al., 2022*). Conversely, stable microbiomes exhibit minimal differentiation between compartments. In *M. auretenra*, coral tissue and mucus microbiomes shared over 50% of taxa, with no significant differences in beta diversity, indicating a conserved, host-associated microbiome with limited external influence.

## The microbiome of *M. auretenra* is predominantly composed of *Endozoicomonas*

We found that the microbiome of *Madracis auretenra* is predominantly composed of Endozoicomonadaceae, Vibrionaceae, and Rhodobacteraceae, consistent with previous studies in other coral species (Table 2). The genus *Endozoicomonas* represented nearly 90% of the total microbial community and, along with *Vibrio*, *Catenococcus*, and *Ruegeria*, constituted the core microbiome of this species. *Endozoicomonas* is generally the most abundant and dominant bacterial taxon in stony corals, often forming aggregates within coral tissues (*Bayer et al., 2013*; *Gotze et al., 2024*; *Maire et al., 2023*; *Wada et al., 2022*). These aggregates can include multiple *Endozoicomonas* phylotypes, which show patterns of co-phylogeny with the coral host and indicate the potential for shared metabolic pathways (*Gotze et al., 2024*; *Hochart et al., 2023*; *Neave et al., 2017*). In our study, 222 unique ASVs were affiliated with *Endozoicomonas*, and we observed that its abundance was influenced by season. High temperatures and total suspended solids (TSS) during the rainy periods were significantly correlated with reduced *Endozoicomonas* abundance. These findings align with observations by *Epstein, Torda & Van Oppen (2019)*; *Pogoreutz et al. (2018)*; *Tandon et al. (2022)*, who reported seasonal effects on Endozoicomonadaceae abundance, highlighting the sensitivity of this taxon to these stressors.

A positive correlation between seasonal phosphate fluctuations and *Endozoicomonas* abundance was observed in our study, marking the first documented evidence of this relationship. Previous studies have shown that this taxon can sequester phosphates within

coral tissues (*Maire et al., 2023*; *Wada et al., 2022*), suggesting its role in mitigating excess phosphorus. This function may be critical in nutrient cycling within eutrophic coastal areas impacted by fertilizer runoff, sewage, and land use changes, where phosphorus overload often drives eutrophication and algal overgrowth (*He et al., 2023*; *Maharjan et al., 2022*). The capacity of *Endozoicomonas* to thrive in nutrient-rich environments, as observed in *Pocillopora verrucosa* unaffected by eutrophication (*Pogoreutz et al., 2018*), underscores its ecological importance in coral reef health. Beyond phosphorus cycling, *Endozoicomonas* contributes to sulfur and carbon cycling through DMSP metabolism (*Tandon et al., 2020*) and regulates the coral microbiome by producing antimicrobials and facilitating essential metabolite exchange (*Neave et al., 2016*; *Ochsenkühn et al., 2023*). The dominance of this taxon in coral *M. auretenra* highlights its multifunctional role in coral health and resilience in nutrient-rich and anthropogenically impacted habitats.

## Stable populations of Vibrionaceae and complex ecological interactions in the *M. auretenra*-associated microbiome

Vibrionaceae was the second most abundant family in the *Madracis auretenra*-associated microbiome, comprising genera such as *Vibrio* and *Photobacterium*. Despite their opportunistic and pathogenic nature, *Vibrio* species display essential roles in coral fitness, including nitrogen fixation in coral mucus (*Chimetto et al., 2008*; *Olson et al., 2009*) and antimicrobial activity against other pathogenic vibrios (*Ritchie, 2006*; *Ruiz-Toquica et al., 2023*), among others. We found *Vibrio* to be a stable component of the core microbiome, with consistent populations across seasons, locations, and stress conditions. While other studies report increased *Vibrio* abundance under coral stress (*Mills et al., 2013*; *Rubio-Portillo et al., 2014*), *Vibrio* levels in *M. auretenra* were unaffected by the coral-algae interaction or in the stressed corals. Although environmental factors like TSS, nitrates, and phosphates were associated with shifts in Vibrionaceae abundance, the changes were not statistically significant, suggesting the presence of unexplored mechanisms regulating *Vibrio* populations within this coral microbiome.

Dominant taxa, such as *Vibrio*, *Ruegeria*, and *Pseudoalteromonas*, may regulate the microbial composition and prevent dysbiosis (*Boilard et al., 2020*; *Clark et al., 2021*). While *Pseudoalteromonas* was barely represented in our study, *Ruegeria* was a core member and has been shown to inhibit pathogenic vibrios during thermal stress (*Miura et al., 2019*) and prevent dysbiosis in bleached corals (*Xu et al., 2024*). Interestingly, despite being considered mutualistic, in the present study, *Ruegeria* was more abundant in corals in contact with algae, suggesting a commensal role. Families like Fusibacteraceae and Rhodobacteraceae, linked to stress in other coral species (*Leite et al., 2018*; *Paulino et al., 2023*), were also detected in stressed *M. auretenra*.

The balance and stability attributes of the *Madracis auretenra* microbiome are supported by complex interactions. The strong co-occurrence between members of Endozoicomonadaceae, Vibrionaceae, and Pseudoalteromonadaceae suggests coexistence, whereas dense connections indicate cooperation and shared niches (functional redundancy) (*Cárdenas et al., 2022*). This cluster likely represents a highly conserved and competitive group, with members exhibiting rapid growth, antimicrobial activity, and nutrient uptake

strategies, all of which contribute to coral health, homeostasis, and functioning (*Bourne, Morrow & Webster, 2016*; *Mohamed et al., 2023*; *Peixoto et al., 2020*; *Voolstra et al., 2024*). The dominance of *Endozoicomonas* and Vibrionaceae (>96%) underpins the stability of this microbiome, offering a unique strategy to cope with environmental variability, such as that present in urbanized coral reefs (*Burt et al., 2020*). The microbiome of *M. auretenra* demonstrates a dynamic balance, supported by a diverse and structurally complex microbial community, where dominant taxa like *Endozoicomonas* and *Vibrio* can contribute to coral health by regulating the microbial composition and preventing the loss of essential functions, a robust strategy for sustaining coral health.

## Nutrients and oxygen levels drive shifts in microbiome composition in *M. auretenra*

Physicochemical variations in the Colombian Caribbean are driven by seasonal changes (*Franco-Herrera, 2005*; *Vega-Sequeda et al., 2019*), geographic factors (*García, Palacio & Garcia, 2012*), and anthropogenic influence (*García-Rentería, Chang Nieto & Cortés, 2023*), all of which shape coral microbiome structure (*Leite et al., 2018*; *Quek et al., 2023*; *Sharp et al., 2017*). Elevated ammonia has been found to alter the *Pocillopora damicornis* microbiome (*Zhang et al., 2021*), whereas higher suspended matter and nutrients during rainy seasons make the *Ctenactis echinata* microbiome more diverse and less structured (*Roder et al., 2015*). Decreased oxygen levels have similarly been shown to negatively affect *Siderastrea siderea* and *Agaricia lamarcki* microbiomes (*Howard et al., 2023*). Similarly, in coral *Madracis auretenra*, shifts in microbiome composition were linked to variations in oxygen, nitrites, nitrates, ammonia, and TSS levels. The seasonal fluctuation in nitrite and oxygen affects the *M. auretenra* microbiome, provoking increases in non-core taxa (*e.g.,* Cyanobiaceae, Actinobacteriota, and Bacteroidota), while ammonia variation influences the core microbiome (Endozoicomonadaceae, Vibrionaceae, and Rhodobacteraceae). The stable abundance of *Endozoicomonas* with nitrates and nitrites variation aligns with *Marchioro et al. (2020)*. The critical roles of this taxon in nitrogen cycling emphasize their importance in maintaining holobiont function and preventing opportunistic invasions triggered by nitrogen excess (*Morris et al., 2019*; *Vanwonterghem & Webster, 2020*; *Wang et al., 2018*). Despite these drivers of shifts in microbial composition, the overall stability of the microbial community structure in *M. auretenra* reflects an adaptive response to environmental stress, including seasonal nutrient fluctuations and anthropogenic impacts. These findings suggest the shifts in microbial composition represent directional adjustments to optimize coral holobiont function, moving it between stable states. Microbial taxa, such as *Endozoicomonas* and Vibrionaceae, play key roles in this phenotypic plasticity, underscoring the dynamic nature of this microbiome and its capacity to adapt to environmental change.

## CONCLUSIONS

The microbiome of *Madracis auretenra* exhibits remarkable stability across seasonal and spatial gradients, with *Endozoicomonas* as the dominant taxon. Environmental factors, including nutrients, oxygen, and suspended solids, significantly influence the microbiome composition, with *Endozoicomonas* showing a positive correlation with phosphate

fluctuations. The dominance of *Endozoicomonas* and the stable presence of Vibrionaceae, along with their complex interactions with other taxa, highlight the functional balance that supports coral health in variable reef habitats. This study emphasizes the critical role of the microbiome in promoting resilience in urban corals, underscoring its importance for coral conservation in highly degraded habitats. Future research should focus on elucidating the functional roles of key microbial taxa in maintaining coral health under changing environmental conditions. Also, long-term monitoring of coral microbiomes across diverse reef types and geographic regions is essential to identifying adaptive strategies and guiding effective management practices.

## ACKNOWLEDGEMENTS

The authors thank the personnel from the Universidad de Bogota Jorge Tadeo Lozano, Santa Marta. We thank the LABCAM unit of the Marine and Coastal Research Institute-INVEMAR for the chemical analysis. We especially thank Luis Garzón for his help in drawing the schematic map, and the staff of the Medina Lab that helped in the preparation of 16S samples for sequencing.

### Funding

This work was supported by the Ministerio de Ciencia y Tecnología (MINCIENCIAS) through the Bicentennial Doctoral Excellence Grant. This study was also funded by NSF grant IOS-2227068 and internal grants from Penn State University. The funders had no role in study design, data collection and analysis, decision to publish, or preparation of the manuscript.

### Grant Disclosures

The following grant information was disclosed by the authors:
Ministerio de Ciencia y Tecnología (MINCIENCIAS).
NSF: IOS-2227068.
Penn State University.

### Competing Interests

The authors declare there are no competing interests.

### Author Contributions

- Jordan Ruiz-Toquica conceived and designed the experiments, performed the experiments, analyzed the data, prepared figures and/or tables, authored or reviewed drafts of the article, and approved the final draft.
- Andrés Franco Herrera analyzed the data, authored or reviewed drafts of the article, and approved the final draft.
- Mónica Medina analyzed the data, authored or reviewed drafts of the article, and approved the final draft.

### Field Study Permissions

The following information was supplied relating to field study approvals (i.e., approving body and any reference numbers):

The collection was performed under permits for the "Collection of Specimens of Wild Species of Biological Diversity for Non-Commercial Scientific Research Purposes". Individual permit expedient PIR 007-2021 was issued through Resolution 191 of 16 December 2021 by the Ministry of Environment and Sustainable Development and National Natural Parks of Colombia. Framework permit was issued through Resolution 1715 of 30 December 2015 and the modified Instruction Normative No. 00213 of 28 January 2021, by the National Environmental Licensing Authority (ANLA) of Colombia.

### Data Availability

The raw sequence data are available at NCBI: PRJNA1103847.

Information about each sample is available in BioSample: SAMN41063004 to SAMN41063075, and the fastaq files are available at NCBI SRA: SRR28786279 to SRR28786350.

The R scripts used for the DADA2 pipeline and to run analysis, statistics, and generate figures, and the filtered table of ASVs, taxonomy, metadata, and environmental data, are available at GitHub and Zenodo:

- https://github.com/JorMicrobe/Madracis_auretenra_Microbiome.git.
- Jordan Ruiz-Toquica. (2025). JorMicrobe/Madracis_auretenra_Microbiome: Coral Madracis auretenra-associated microbiome (Coral_microbiome). Zenodo. https://doi.org/10.5281/zenodo.15006550

### Supplemental Information

Supplemental information for this article can be found online at http://dx.doi.org/10.7717/peerj.19226#supplemental-information.

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
