# Peer review of "Endozoicomonas dominance and Vibrionaceae stability underpin resilience in urban coral Madracis auretenra"

_PeerJ, doi:10.7717/peerj.19226_

## Round 0.1 · original submission · Major Revisions

We have concluded the revision of your manuscript. Two expert reviewers have evaluated your manuscript and as you can see below, both have undertaken a thorough revision and have provided excellent and detailed comments for each section of your manuscript. Due to the extent of the revisions required I have decided that a major revision is required. As you prepare the revised version of your manuscript, please include a careful accounting of every suggestion and how (and where) it was incorporated or otherwise addressed in the manuscript so that the reviewers and I have the easiest time verifying the improvements and clarifications.

Reviewer 1 ·

Basic reporting

Clear English used throughout, sufficient references and background. Article structure is good although Introduction and Discussion sections need to be more concise. Raw data has been fully shared and sequences will be released timely.
Results are relevant but the authors present a complex work, which makes the results difficult to understand and interpret. Given their complexity, it would be better if the information is synthesized, highlighting the main variations found. Some contrasting interpretations and multiple analyses performed, all described in detail, make the text hard to follow. Using fewer words generally improves clarity.
The presentation of results needs much work to help the reader identify the main points supported. For example, when two analyses are used (PERMANOVA and paired comparisons) and the results are similar, rather than describing them in both analyses, it would be better to simply state that the results were the same. Another example: lines 411-414 give information that can be obtained from the figure (3), so I would consider referring to the information without describing it in depth.
The information given on Table 1 is re-described in the text (lines 505-514); I think this is unnecessary, since an analysis of the information follows (Permanova and Anosim results). It would be better to highlight important changes in any case.
In general, I expected P-values to be given, rather than “P < 0.05” or “P > 0.05”.
The figures presented are sufficient to illustrate the work done; even tough, some are very complex (e.g. Fig 4). However, figures need some corrections (see below).

Experimental design

The research is of interest to this journal and the investigation was performed with high technical and ethical standards. Methods are described in sufficient detail.
The research question is well defined and relevant. However, the authors can improve their manuscript by explaining simply and clearly how this work fills a knowledge gap, and how it contributes to the general knowledge of coral microbiomes.

Validity of the findings

The authors should construct a discussion with main findings that contribute to support or refute their own. Sometimes there is information not directly related to their results.
In general, the subtitles (Results and Discussion) are too long and complex. The authors should choose the important facts that the results support.
In the discussion, some statements are hard to support. For example, a stable microbiome does not necessarily share a significant percentage of taxa with the surrounding seawater. I would argue that seawater bacterioplancton and coral microbiomes are not similar and as such, cannot be used as an argument to propose a highly stable microbiome.
Conclusions are sufficiently stated.
I find this work publishable but needing mayor revisions.

Additional comments

Specific comments:
Line 62: replace “between” with “among”
Line 85: revise, orphan phrase? “…the microbiome may experience shifts in the composition (Rajasabapathy…)”
Line 108: please define “urban corals”
Line 187: how did you keep samples at 4°C on the boat?
Line 200: centrifuged for how long?
Line 224: how did you assess “pure DNA”?
Line 242: which database did you use? Silva 132 or 138?
Line 264: I am not sure what you mean by compartments, please clarify
Line 359: I suggest shortening the subtitle in this section, the different diversity between corals and seawater is not new information
Lines 367, 368, 369: please give the actual P-values
Line 370-371: what do you mean by “adjusted p<0.05, or p>0.05”? (Also lines 375 & 376)
Lines 390-391: In the figure it is not clear which taxa are “unique”; as suppose as opposed to “shared”? Then, I suggest using “not shared” rather than “unique”
Line 394: I would replace “compositional structure” with “beta diversity”
Line 400: I am not sure how is this statement different from the one given above (lines 398-400)
Line 404: it is hard to judge if similarities within and between groups were greater or not if no information on P-values is given
Line 406: do you mean among groups?
Lines 495-496: remove phrase “This family…” is not significant
Lines 514-521: The PCA analysis, to my understanding, identified as the most variable parameters, temperature, salinity, and nitrates.
Lines 549-550: I suggest eliminating the phrase “Further, some Endozoicomonadaceae…
Line 555: I am not sure if you mean that members of Endozoicomonadaceae are not present when Alteromonadaceae is? Because in the same paragraph you found a big cluster with these two families (lines 547-548).
Line 567: “between in two”, please correct
Lines 566-569: I would erase this phrase (or make it shorter); go directly to your findings
Lines 581-584: I think you should be more cautious with this statement, you only looked at two consecutive years
Lines 602-605: I find this discussion out of place. Should be closer to the end of this section
Line 624: replace “Contrasting” with “contrastingly”
Lines 645-648: please revise
Line 658: “consistent with various studies”, but in the same coral species or others, please revise
Line 671: “affected by the weather” replace with “affected in the rainy season”
Line 674: “low temperature” you mean “lower”
Lines 696-697: remove phrase, it is not pertinent to your data
Line 723: replace “not assigned” with “unassigned”
Lines 746-753: I am not sure of its relevance here
Line 774: “homogeneous microbiome” needs to elaborate further
Lines 783-785: I think this is stretching too much your results
Lines 785-790: This text seems unnecessary, please revise
Line 803: What do you mean by “key evidence”, as opposed to just evidence, please explain
Line 810: Can you please elaborate a bit regarding microbiome reshape vs. restructure (superficial vs. profound changes?) or use different words?

Figure 2 = needs color or some other way to distinguish among ASV data; the figures separate only shared ASVs by color, not unique or ubiquitous ones, as the title claims. To my understanding “unique” taxa is not the same as “not shared”. Please revise.
The caption to Fig 2 should state that the numbers given are for P-values after comparing the diversity measures with the Kruskal-Wallis test.

Figure 3 = please give actual P-values in the caption. Symbols for Protected and Urban sites are too small. Also, for Fig 3B, I recommend giving a more contrasting palette of colors to distinguish the different families with ease.

Figure 4 = the text “of the physicochemical variables” is repeated, as well as “nitrites”. The term “Perms” is not in Fig 4A.
To my understanding, PCA plots identify main components that contribute to the variation. In your plot (A), PC1 incorporates temperature, salinity and nitrites, that varied the most. PC2 incorporates TSS, phosphates, oxygen and ammonia. Other variables (pH, nitrates) have a variation in both components. On the other hand, CCA helps find correlations among variables. For example, oxygen and mucus seem correlated, also ammonia and nitrates with tissues.
In Fig4 C it is hard to appreciate anything, the colors do not help.
I would suggest moving Fig4 E to supplementary material.

·

Basic reporting

General Comment:

This work is recommended for publication, following the suggested revisions and improvements:

The paper provides a detailed analysis of the stability and composition of the microbiome in M. auretenra, highlighting the relevance of specific taxa such as Endozoicomonas and Vibrionaceae in coral health and their response to environmental factors. The study reveals a remarkable stability in the microbiome over time and space, even under varying environmental conditions, which is a significant finding for coral ecology and marine microbiology. Additionally, the study emphasizes how seasonal factors affect the abundance of certain taxa, as well as the variability in microbial diversity among different compartments of the coral. These findings are crucial for understanding the mechanisms underlying coral health and resilience in the face of environmental changes.

Given that this work provides novel and relevant insights into coral microbiome stability and its response to environmental factors, and significantly contributes to the fields of coral ecology and marine microbiology.

Important about References list: Due to the very extensive list of cited references, conduct a thorough and critical review of the references, and only keep the most relevant ones to support your study.

Experimental design

Suggested Changes:

• INTRODUCTION:
The introduction thoroughly covers the challenges faced by coral reefs, focusing on the role of the microbiome in coral resilience. There's an extensive use of references, showcasing the depth of research and providing a solid foundation for the study. The text clearly articulates the importance of studying coral microbiomes, especially in urban-influenced environments, which is a unique and significant aspect.

Clarity and Conciseness: The introduction is dense and covers multiple topics. Streamlining some sections could help maintain focus and readability. Some sections could benefit from smoother transitions to guide the reader through the different themes more effectively. However, the hypothesis is mentioned towards the end, but a more explicit statement of the research objectives earlier in the introduction might help guide the reader. While the introduction mentions urban corals, it could benefit from emphasizing the uniqueness and relevance of studying these specific coral types earlier in the text.

Streamline and Clarify:
• Consider breaking down complex sentences into shorter, clearer ones to enhance readability.
• For example, the sentence: "These stressors are decimating coral populations and leading to the decline of reefs (Hoegh-Guldberg & Bruno, 2010; Bellantuono et al., 2012; Brown et al., 2017; Hoegh-Guldberg et al., 2017; Eddy et al., 2021)" could be simplified to: "These stressors are leading to significant declines in coral populations and reef ecosystems."

Early Hypothesis Statement:
• Introduce the hypothesis earlier. After discussing the resilience of urban corals, you could add: "We hypothesize that the unique microbial communities in urban corals play a critical role in their resilience to environmental stressors."

Focus on Urban Corals:
• Emphasize the significance of urban corals in the study early on. For instance, after discussing general coral challenges, you could state: "Among these, urban coral reefs, often exposed to higher anthropogenic impacts, present a unique opportunity to study how coral-associated microbiomes contribute to their survival and resilience."


Questions to Consider
Research Focus: How do the microbiomes of urban corals differ from those in less impacted areas? What specific factors contribute to these differences?

• Methods:

Clarification:
- What steps were taken to minimize cross-contamination during sample collection?
- How did you ensure that the results were not biased by contamination during sample collection and analysis?
- Did you implement appropriate controls to ensure data quality?
- Could the lack of significant differences in compositional structure between stations be due to the sampling frequency or the time period analyzed?
- Would it be useful to conduct a longer-term follow-up?

Validity of the findings

Discussion:

General Comment:Your discussion is comprehensive and presents a clear understanding of the spatio-temporal dynamics of coral microbiomes. To refine and deepen the analysis, here are a few points of consideration and potential questions that could enhance the discussion:

Questions to Consider in Discussion section:

Complexity of Microbiome Stability:

- Question: How do you reconcile the observed stability in the M. auretenra microbiome with studies that report significant microbiome shifts under environmental stress? Are there specific environmental factors or thresholds that might lead to microbiome instability?
- Consideration: Discuss potential mechanisms that M. auretenra might use to maintain microbiome stability. Is there evidence that certain microbial taxa have adaptive mechanisms that contribute to this stability?

Impact of Environmental Variables:

- Question: While seasonal fluctuations and localities are identified as primary drivers of microbiome changes, are there other subtle environmental or biological factors (e.g., microhabitat variations, coral reproduction cycles) that could influence the microbiome dynamics?
- Consideration: It might be useful to explore whether there are interactions between different environmental variables that collectively impact the microbiome in a more nuanced way than single-factor analyses suggest.

Role of Endozoicomonas and Vibrionaceae:

- Question: How does the seasonal variation in phosphate concentration influence the functional roles of Endozoicomonas in coral health? Are there any functional assays or metabolic pathways that could provide further insight into this relationship?
- Consideration: Given the potential role of Vibrionaceae in coral health, could there be species-specific interactions within this family that contribute to the stability of M. auretenra's microbiome? It might be beneficial to investigate the functional diversity within Vibrionaceae.

Microbiome Reshaping vs. Restructuring:

- Question: Could the observed stability in the microbiome be a result of a balance between microbiome reshaping and restructuring? What are the specific indicators or thresholds that distinguish between these processes?
- Consideration: Explore how different environmental pressures (e.g., temperature, nutrient levels) might trigger reshaping versus restructuring of the microbiome. Are there specific conditions under which one process is favored over the other?

Comparative Analysis with Other Coral Species:

- Question: How do the findings for M. auretenra compare with other coral species in similar or different environments? Are there universal patterns in microbiome stability and flexibility across coral taxa?
- Consideration: Compare and contrast with studies on other corals with stable or flexible microbiomes. Are there shared characteristics or conditions that contribute to these patterns?

Microbial Interactions and Network Analysis:

- Question: The ecological network suggests co-existence among certain taxa. How do these interactions influence the overall microbiome stability and coral health? Are there any key microbial interactions that might be crucial for maintaining stability?
- Consideration: Delve into the network analysis to identify key microbial interactions. How might these interactions contribute to the resilience or health of the coral holobiont?

Addressing these questions and considerations could provide deeper insights into the stability and dynamics of the coral microbiome, potentially revealing more about the underlying ecological and evolutionary processes.

Additional comments

Review the grammar in detail:

L45: "These stressors are decimating coral populations and contributing to the decline of reefs..."
L51: "...exhibit complex spatio-temporal dynamics shaped by multiple factors."
L52: "Although similar microbial communities can be observed across coral species..."
L56: "These characteristics depend on various factors, including (i) the algae-coral symbiosis..."
L59: "The study of the influence of these characteristics on the coral microbiome has grown significantly over the last decade."
L63: "Symbiodiniaceae provide essential nutrients to the coral host through photosynthesis..."
L69: "However, the abundance and composition of these microbes vary depending on factors such as the host species..."
L71: "...geographic location, environmental conditions (e.g., temperature, light intensity, and nutrient availability)..."
L79: "...the proximity to urban settlements, and pollution, shape less homogeneous microbiomes..."
L86: The increase in the abundance of → An increase in the abundance of
L90: Bleached corals exhibit → Bleached corals often exhibit
L 92: and for → and to
L94: are represented by → is represented by
L 96: are, making these hosts and environments → become, making hosts and environments
L97-98 or to switching → or to the switching of
L98-99: This reduces → These reduce
L101: which include → that include
L105: and leads to → leading to
L106: Studying unique host-associated microbiomes → Studying these unique host-associated microbiome
L108: Urban corals are attractive subjects → Urban corals are particularly attractive subjects
L109-110: essential in their resilience → that are essential to their resilience
L111: shape → adapt
L111: and enhance survival → and to enhance survival
L113: suggesting roles → suggesting their roles
L117: that could be related to→which could be related to
L118-119: In the Colombian Caribbean, coral species have adapted → In the Colombian Caribbean, several coral species have adapted
L120: Varadero and Rosario reefs → the Varadero and Rosario reefs
L122: Other study in the Tayrona National Natural Park → Another study in the Tayrona National Natural Park
L124: suggesting control mechanisms → suggesting the existence of control mechanisms
L124: that are specific of each holobiont → specific to each holobiont
L151: Resolution 191 of 16 December 2021 → Resolution 191 on December 16, 2021
L154: 28 January 2021, → January 28, 2021,
L159: high signs of deterioration → significant deterioration
L167: if → whether
L172: whole study → entire study
L178: deterioration sign → signs of deterioration
L202: Syringe Filters Millex®-GS → Millex®-GS Syringe Filters
L204: through the DNeasy® Powersoil® Pro Kit → using the DNeasy® Powersoil® Pro Kit
L209: mins → min
L212via Nanodrop Multiskan™ Go Microplate → using a Nanodrop Multiskan™ Go Microplate
L214: on triplicates, → in triplicates,
L219: The 16S rRNA V4 region was amplified using CS1_515F and CS2_806R from Roitman et al. (2020), with linker sequences and Illumina adapters (Caporaso et al., 2012; Roitman et al., 2020).
L223-227: 50 ng of pure, high molecular weight DNA was used with a master mix containing: 5 µL PCR Buffer II 10X, 2.5 µL dNTPs mix, 2.5 µL forward primer, 2.5 µL reverse primer, 1.5 µL MgCl₂, 0.2 µL AccuStart II™ Taq Polymerase, and sterile water.
L237-238: Reads were truncated at positions 220 (forward) and 200 (reverse) where quality dropped.
Line 242: Chimeric sequences were removed, and taxonomy was assigned using SILVA SSU data.
L360: n=72 → (n = 72).
L362: was ranging from → ranged from.
L363: enough sequencing of each sample that covered the microbial diversity → sufficient sequencing depth to cover microbial diversity.
L477-78: significant changes → significant differences.
L484: showed no differential abundance among → did not vary significantly among.

L487: however, this taxon was not differentially abundant → however, this taxon did not show significant differences.
L 490: was significant → was significantly different.
L491: during the rainy seasons → during rainy seasons.
L496: but this was not significant → although this was not significant
L512: TSS was reported → TSS was observed.
L514-15: rather than local → rather than local factors.
L515-16: This was confirmed in → This was supported by.
L518: (>24% contribution) → (>24% of contribution).
L541: significantly correlated → significantly associated.
L553-54: Endozoicomonadaceae, Vibrionaceae, and Pseudoalteromonadaceae… → seem Endozoicomonadaceae, Vibrionaceae, and Pseudoalteromonadaceae. In contrast, other members appear to be mutually
L571: suggesting this stability is key → suggesting that this stability is key.
L673: high phosphates → high phosphate levels.
L630: the concentration of phosphates and the abundance → the concentration of phosphates with the abundance.
L704-705: Vibrio spp. have been rarely considered mutualistic bacteria → Vibrio spp. are rarely considered mutualistic bacteria".
L746: Coral and seawater share several microbial taxa → Corals and seawater share several microbial taxa.
Línea 740-41: has been reported as made up of primarily opportunistic bacteria → has been reported to be composed primarily of opportunistic bacteria

Grammatic:

Table S13. Selection of the most representative variables for the canonical correspondence analysis (CCA). The variance inflation detects the linear dependency of the variables, values <10 indicates non collinearity. CCA1 and CCA2 components, R2 value and the statistic test to detect significant variable in the ordination 

Table S13. Selection of the most representative variables for the canonical correspondence analysis (CCA). The variance inflation detects the linear dependency of the variables, values <10 indicate non-collinearity. CCA1 and CCA2 components, R2 value, and the statistic test to detect significant variables in the ordination

- Figure 3 C: Try changing the color scale; the light yellow and white tones in the Phylum Firmicutes are lost.

---

## Round 0.2 · accepted · Accept

I have received evaluations of your resubmission and am delighted to let you know that your manuscript is now suitable for publication in PeerJ. I sincerely appreciate that you took the time to attend to all of the comments made by the reviewers. Congratulations.

Reviewer 1 ·

Basic reporting

Clear writing and good structure. Sufficient references used correctly in context. Figures included are relevant, contributing to a self-contained study.

Experimental design

This is an original contribution, with a research question well defined; the methodology is detailed and well explained.

Validity of the findings

The discussion has been rewritten to present the main findings clearly. The information presented is of great interest, highlighting resilient corals that grow in urban sites, yet having a stable microbial community. The dominance of Endozoicomonas is also observed in these urban reefs.

Additional comments

The authors have done a great job in the resubmitted manuscript. Suggestions by reviewers have been considered and now the ms. is more concise, better structured, easy to read and follow, and highlights the main findings of the study.
I think the manuscript is ready for acceptance.

·

Basic reporting

The manuscript has been carefully reviewed and meets the scientific and editorial standards of PeerJ. The suggested revisions have been successfully incorporated, resulting in a clear and well-structured document with solid methodological and bibliographic support. The tables and figures have been improved for better clarity, enhancing the presentation of the results. Additionally, the revised title more accurately reflects the study’s content and scope.

The writing is professional, clear, and appropriate for a scientific journal, with no major grammatical errors. The manuscript includes relevant and up-to-date references, following PeerJ’s guidelines. The figures and tables are well-organized, and the availability of raw data in public databases ensures compliance with the journal’s requirements.

Experimental design

The research question is clear and relevant, with an improved introduction that follows previous recommendations and highlights the importance of studying microbiomes in urban corals. The methodology has been refined for greater clarity and reproducibility, providing a more precise description of data processing and the controls implemented to prevent contamination and sampling biases.

Validity of the findings

The statistical analysis is robust, and the results are consistent. Following previous recommendations, the conclusions have been strengthened and are well-supported, effectively linking to the research question and obtained results. The key findings are adequately summarized, highlighting the importance of stable microbiomes in urban corals.

Additional comments

Since the manuscript has been refined based on previous recommendations and does not present significant deficiencies, I consider it ready for publication in its current version.